# Osteogenic Potential of Nano-Hydroxyapatite and Strontium-Substituted Nano-Hydroxyapatite

**DOI:** 10.3390/nano13121881

**Published:** 2023-06-17

**Authors:** Georgia-Ioanna Kontogianni, Catarina Coelho, Rémy Gauthier, Sonia Fiorilli, Paulo Quadros, Chiara Vitale-Brovarone, Maria Chatzinikolaidou

**Affiliations:** 1Department of Materials Science and Technology, University of Crete, 70013 Heraklion, Greece; gi.kontogianni@gmail.com; 2FLUIDINOVA S.A., 4475-188 Maia, Portugal; catarina.coelho@fluidinova.pt (C.C.); paulo.quadros@fluidinova.com (P.Q.); 3Department of Applied Science and Technology, Politecnico di Torino, 10129 Turin, Italy; remy.gauthier@cnrs.fr (R.G.); sonia.fiorilli@polito.it (S.F.); chiara.vitalebrovarone@polito.it (C.V.-B.); 4CNRS, INSA Lyon, Université Claude Bernard Lyon 1, UMR 5510, MATEIS, F-69621 Villeur-banne, France; 5Foundation for Research and Technology Hellas (FORTH), Institute for Electronic Structure and Laser (IESL), 70013 Heraklion, Greece

**Keywords:** bone regeneration, osteoinduction, cytotoxicity, osteogenic differentiation, pre-osteoblasts, RT-qPCR

## Abstract

Nanohydroxyapatite (nanoHA) is the major mineral component of bone. It is highly biocompatible, osteoconductive, and forms strong bonds with native bone, making it an excellent material for bone regeneration. However, enhanced mechanical properties and biological activity for nanoHA can be achieved through enrichment with strontium ions. Here, nanoHA and nanoHA with a substitution degree of 50 and 100% of calcium with strontium ions (Sr-nanoHA_50 and Sr-nanoHA_100, respectively) were produced via wet chemical precipitation using calcium, strontium, and phosphorous salts as starting materials. The materials were evaluated for their cytotoxicity and osteogenic potential in direct contact with MC3T3-E1 pre-osteoblastic cells. All three nanoHA-based materials were cytocompatible, featured needle-shaped nanocrystals, and had enhanced osteogenic activity in vitro. The Sr-nanoHA_100 indicated a significant increase in the alkaline phosphatase activity at day 14 compared to the control. All three compositions revealed significantly higher calcium and collagen production up to 21 days in culture compared to the control. Gene expression analysis exhibited, for all three nanoHA compositions, a significant upregulation of osteonectin and osteocalcin on day 14 and of osteopontin on day 7 compared to the control. The highest osteocalcin levels were found for both Sr-substituted compounds on day 14. These results demonstrate the great osteoinductive potential of the produced compounds, which can be exploited to treat bone disease.

## 1. Introduction

The native human bone extracellular matrix (ECM) is made from both inorganic and organic compounds [1], with hydroxyapatite (Ca_10_(PO_4_)_6_(OH)_2_, HA) being the most abundant inorganic component in the form of nanocrystals that align with collagen nanofibers. This unique nanostructure, along with its chemical composition, serves a dual purpose, acting as a hard material providing mechanical support to the body and protecting vital organs, and maintaining calcium homeostasis [2]. Additionally, the large surface area of HA nanocrystals facilitates their rapid dissolution by osteoclasts, contributing to the maintenance of calcium ions in body fluids and promoting appropriate bone remodeling rates [3]. Due to these properties, synthetic hydroxyapatite has been used for many decades in the field of biomedical applications due to its similarity to native HA-containing bone [4]. This similarity results in high biocompatibility and osteogenic properties [5]. However, the poor mechanical strength of pure hydroxyapatite scaffolds limits their clinical applications in implants [6]. The development of nanohydroxyapatite (nanoHA) materials that mimic the native bone hydroxyapatite nanocrystals further enhances the properties of HA and the application of nanoHA for bone regeneration in vitro and in vivo [7]. Specifically, nanoHA holds great promise for various applications, particularly in the treatment of bone defects such as those derived from osteoporotic pathologies, a significant global health concern characterized by brittle bones [8], leading to decreased bone mass, increased fracture risk, and reduced bone strength. In this context, bone fractures can be treated either by filling them with medical devices enriched with nanoHA, such as bioactive bone cements and osteogenic injectable hydrogels and microspheres [9] containing nanoHA, polymeric composite matrices such as 3D osteogenic scaffolds [10], or coatings of metallic implants with nanoHA compounds [11,12]. The use of minimally and noninvasive therapeutic agents for bone tumor ablation and bone regeneration as bifunctional materials is another recently reported challenge [13]. The incorporation of nanoHA particles into different materials can improve the material properties. For instance, the incorporation of 50 wt% nanoHA in poly(L-lactic acid) 3D-printed scaffolds increased the Young’s modulus, which reached values close to the modulus of the human trabecular bone [14]. Another report showed that the addition of nanoHA in alginate hydrogels reinforced the physicochemical properties of the material and enhanced the osteogenic potential in vitro and ex vivo [15]. Silk fibroin/nanoHA hydrogels have been reported to promote human bone marrow stromal cell proliferation and differentiation towards the osteoblastic lineage [16]. Moreover, HA nanoparticles have been used in oral care products, including toothpastes, mouthwashes, and dental items, to reduce dentine hypersensitivity [17,18]. Furthermore, nanoHA has been described as a promising material for drug delivery in cancer treatment [19], rheumatoid arthritis [20], and bone infections [21].

However, pure nanoHA materials still lack inherent osteoinduction and, thus, limit the formation of new bone [22]. In order to overcome this limitation, recent research attention increasingly focuses on native HA crystals that contain different ions, including strontium (Sr) among others. The incorporation of different ions in nanoHA crystals via the replacement of calcium (Ca) ions may positively influence the osteogenic properties of the materials and their biocompatibility [23], more closely mimicking the mineral phase of native bone tissue.

Sr is a widely used trace element for bone tissue regeneration [24], including the treatment of osteoporotic fractures, due to its ability to enhance osteogenesis [25,26,27,28]. The key aspect that led the scientific community to use Sr in bone tissue engineering applications is its chemical analogy to Ca, since both are divalent cations with similar ionic radii [29]. Based on this fact, studies with the use of Sr isotopes (Sr-83) show that Ca and Sr are stored similarly in the human organism with few biological differences [30]. In 2021, Borciani et al. [24] reviewed different types of biomaterials for bone tissue engineering incorporating Sr, with particular reference to the effect of this ion on mesenchymal stem cells, osteoblasts, and osteoclasts. The incorporation of 5% Sr in sol–gel-produced biphasic calcium phosphate (BCP) powders showed enhanced osteogenic properties with increased synthesis of collagen type I in human primary bone cells [31]. In another study, silk fibroin/carboxymethyl chitosan with Sr-substituted HA showed enhanced alkaline phosphatase (ALP) activity and higher expression levels of runt-related transcription factor 2 (RUNX2), ALP, osteocalcin (OCN), osteopontin (OPN), bone sialoprotein (BSP), and collagen type I compared to the control materials without Sr [32]. Sr seems to have a dual role in bone regeneration process. Beside its effect to promote osteogenesis, the osteoclastogenic effect of Sr has also been described. It has been shown that it can reduce the osteoclastic activity in a dose-dependent manner, causing effects ranging from an alteration of the osteoclast cytoskeleton to a total inhibition of osteoclast formation [33,34]. Thus, the substitution of Ca with Sr serves as an attractive method to produce compounds to treat bone defects caused by bone fragility related to excessive osteoclastic activity and the suppression of osteoblastic activity, a situation faced in osteoporosis. Sr substitution in calcium phosphates has been studied in coculture models of osteoblasts and osteoclasts promoting the regenerative ability in vitro and in vivo [35]. Moreover, besides the favorable biological properties, the incorporation of Sr in HA crystals has been described to improve the mechanical properties of HA materials [36]. Substitution with a Sr concentration of 10 mol% enhances the dissolution kinetics of nanoHA and, thus, its degradation [37,38]. Even though the incorporation of Sr into nanoHA crystals enhances their properties, there are only a few reports on concentrations higher than 10 mol% [39,40]. The method of production is an important aspect to consider in Sr substitution. A research study on the production of two Sr-substituted nanoHA materials, at 25 and 50 mol%, in the form of powder using spray drying [39] reported high cytotoxicity in direct contact with L929 fibroblasts at a concentration of 100 μg/mL. In contrast, substitution degrees of 0, 2.5, 5, 10, 50, and 100 mol% of Sr-substituted nanoHA pastes produced via rapid-mixing wet precipitation showed low cytotoxicity when 0.1 mL of each paste was cultured in direct contact with MG63 cells [40]. In the latter study, the authors did not report on the differentiation capacity of the synthesized materials. There is still a need for developing compounds with increased Sr substitution degrees into nanoHA crystals with enhanced osteogenic properties without cytotoxic effects.

In this study, nanoHA was synthesized and substituted with 50% and 100% Sr (Sr-nanoHA_50 and Sr-nanoHA_100, respectively). These materials were produced through wet chemical precipitation using Ca, Sr, and phosphorous salts as the starting materials. The nanoHA materials with or without the incorporation of Sr were diluted in culture medium at a concentration of 0.25 %*v*/*v*, which corresponds to 37.5 mg/mL of Ca_(10−x)_Sr_x_(PO_4_)_3_(OH), and they were evaluated for their biocompatibility in vitro, in direct contact with MC3T3-E1 pre-osteoblastic cells. The osteogenic potential of the nanoHA materials was evaluated in terms of the determination of the ALP activity, the calcium concentration, and the collagen production. Moreover, the gene expression levels of the osteogenesis-related markers ALP, osteonectin, osteopontin, and osteocalcin were assessed by means of a real-time quantitative polymerase chain reaction (RT-qPCR). 

## 2. Materials and Methods

### 2.1. Preparation of Sr-Substituted Hydroxyapatite Materials

The nanoHA materials were prepared according to an existing protocol using the NETmix^®^ reactor [41,42]. Briefly, aqueous solutions of Ca^2+^ and PO_4_^3−^ ions were prepared and fed into the reactor under controlled pH conditions. To obtain the Sr-nanoHA materials, a source of Sr^2+^ was also added. The source for Sr^2+^ was SrCl_2_·6H_2_O. The synthesis of the Sr-nanoHA consisted of feeding a solution containing calcium and/or strontium and a potassium phosphate solution. The samples were obtained using wet chemical precipitation and the resulting slurries were continuously washed to remove residual ions and further concentrated to obtain suspensions with a final concentration of 15 ± 1.0 wt%, with 100% of Ca^2+^ in nano-HA, 100% of Sr^2+^ substitution in Sr-nanoHA_100, and 50% of Sr^2+^ substitution in the case of Sr-nanoHA_50. Table 1 lists all the materials produced (Ca_(10−x)_Sr_x_(PO_4_)_3_(OH)_2_, x = 0, 5 or 10). 

### 2.2. Characterisation of the Nano-HA Particles

The morphology of the obtained nanoHA materials was analyzed through field-emission scanning electron microcopy (FE-SEM, ZEISS MERLIN instrument, Oberkochen, Germany). For this purpose, suspensions of nanoHA, Sr-nanoHA_50, and Sr-nanoHA_100 powders in isopropanol were prepared, dropped on a copper grid (3.05 mm, diam.200 MESH, TAAB, Aldermaston, Berks, UK), and left at room temperature to allow for isopropanol evaporation. A chromium layer was then sputtered on the dried samples prior to the analysis. The amount of incorporated Sr and the (Ca + Sr)/P ratio were evaluated through energy-dispersive X-ray spectroscopy (EDS) using a desktop SEM (Desktop SEM Phenom XL, Phenom-World B.V., Netherlands). 

X-ray diffraction (XRD, X’Pert PRO, PANanalytical, Almelo, The Netherlands) measurements were performed with Cu_Kα_ radiation. The instrument operated at 40 kV and 40 mA with a 0.013 2θ step between 10° and 80° and a scan step time of 45 s. The XRD diffractograms were analyzed using X’Pert HighScore software. The Scherrer equation was used to estimate the crystallite size (nm) based on the broadening of the (002) XRD peak according to the following Equation (1):(1)$$L=\frac{K\lambda }{\beta cos(\theta )}$$
where *L* = crystallite size, *K* = 0.9 is the Scherrer constant, *λ* = 1.5406 Å is the X-ray beam wavelength, β is the 002 peak width at half the maximum intensity (radians), and *θ* is the Bragg angle. 

Finally, the amount of Sr ion released from Sr-nanoHA_50 and Sr-nanoHA_100 upon their dissolution was analyzed through the inductively coupled plasma atomic emission spectrometry technique (ICP-AES) (ICP-MS, Thermo Scientific, ICAP Q) following the protocol optimized in [43]. Briefly, 5 mg of powders was suspended in 20 mL of Tris HCL buffer (Tris(hydroxymethyl) aminomethane (Trizma) (Sigma Aldrich, Darmstadt, Germany) 0.1 M, pH 7.4) and kept in an orbital shaker (Excella E24, Eppendorf) with an agitation rate of 150 rpm. The Sr release was followed over 21 days and monitored at the following time points: 3 h, 24 h, 3 days, 7 days, and 14 days. At each time point, the samples were centrifuged at 10,000 rpm for 5 min (Hermle Labortechnik Z326, Wehingen, Germany) and half the supernatant was collected and analyzed through ICP-MS. The volume of the collected supernatant was replaced with the same volume of fresh Tris HCL solution.

### 2.3. Culture of MC3T3-E1 Pre-Osteoblastic Cells

A well-established cell line was used for the evaluation of the cytotoxicity and the osteogenic potential in direct contact with the materials. Pre-osteoblastic MC3T3-E1 cells (DSMZ Braunschweig, Germany, ACC-210) from newborn mouse calvaria were used. These cells exhibit an osteoblastic phenotype and have the potential to differentiate into osteoblasts [44]. The cells were cultured in α-MEM medium supplemented with 10% (*v*/*v*) fetal bovine serum (FBS) (PAN-Biotech, Aidenbach, Germany), 100 μg/mL penicillin and streptomycin (PAN-Biotech, Aidenbach, Germany), 2 mM L-glutamine (PAN-Biotech, Aidenbach, Germany), and 2.5 μg/mL amphotericin (Thermo Fisher Scientific, Waltham, MA, USA) in a humidified incubator at 37 °C, 5% CO_2_ (Heal Force, Shanghai, China). The cell culture medium was replaced twice weekly and confluent cells were detached using trypsin-0.25% ethylenediaminetetraacetic acid (EDTA) (Gibco, Thermo Fisher Scientific, Waltham, MA, USA). MC3T3-E1 cells between passages P14-16 were used for the experiments.

### 2.4. Cell Seeding and Culture in Direct Contact with Hydroxyapatite Materials

For the biological experiments, each of the nanoHA materials was immersed in complete α-MEM medium at a concentration of 0.25% (*v*/*v*), which corresponds to 37.5 mg/mL of Ca_(10−x)_Sr_x_(PO_4_)_3_(OH)_2_. This concentration was selected based on our own data on the cytotoxicity of serial concentrations indicating that the concentration of 0.25% (*v*/*v*) is not cytotoxic. For the cell viability assessment and the differentiation experiments, the materials were diluted in complete osteogenic medium, containing 10 mM β-glycerophosphate, 50 μg/mL l-ascorbic acid, and 10 nM dexamethasone (all provided by Sigma, Darmstadt, Germany). For the live/dead assay, complete medium without the addition of osteogenesis factors was used. 

For the cell viability quantification by means of the PrestoBlue^TM^ assay and the differentiation experiments (biochemical analysis), 5 × 10^4^ cells/well were seeded in 24-well plates and for the live/dead assay, 5 × 10^3^ cells/well were seeded in 24-well plates. For the gene expression experiments by means of the real-time quantitative polymerase chain reaction, 15 × 10^4^ cells/well were seeded. Tissue culture polystyrene (TCPS) was used as control culture for all the experiments. For each experiment, the cells were seeded and left to adhere for 24 h at 37 °C, and the materials were added the next day.

### 2.5. Biocompatibility Assessment of Hydroxyapatite Materials

The biocompatibility assessment of nanoHA materials and TCPS control was performed using a PrestoBlue^TM^ (Invitrogen Life Technologies, Waltham, MA, USA) assay. This method is based on the ability of live cells to retain a reducing environment in their cytosol. PrestoBlue^TM^ is a resazurin-based indicator, which live cells can uptake and reduce to a red product that can be detected spectrophotometrically. Briefly, at each time point tested, the PrestoBlue^TM^ was added to the wells (1:10 diluted in complete medium) and incubated at 37 °C for 1 h. In order to measure the product without the interference of the materials and the cells, aliquots of 100 μL were transferred to a 96-well plate and the absorbance (570/600 nm) was measured using a spectrophotometer (Synergy HTX Multi-Mode Microplate Reader, BioTek, Winooski, VT, USA). The absorbance values were translated to cell numbers by means of a calibration curve of known cell numbers per well. Quadruplicates of three independent experiments (*n* = 12) were analyzed.

### 2.6. Adhesion and Morphology of Pre-Osteoblastic Cells in Direct Contact with the Hydroxyapatite Materials

The cell viability of nanoHA was further evaluated using a live/dead assay (Biotium, Fremont, CA, USA), according to the manufacturer’s instructions. Briefly, after 3, 7, and 14 days of culture, the cells were washed with PBS and then incubated with a working solution of 2 mM calcein-AM (494 nm excitation/517 nm emission) and 4 mM ethidium homodimer III (EthDIII) (532 nm excitation/625 nm emission) at room temperature for 45 min. The cells were washed with PBS and observed under an inverted fluorescence microscope. Visualization of the fluorescently labelled cells was performed using the ImageJ software (ImageJ software, sLOCI, University of Wisconsin, Madison, WI, USA). 

### 2.7. Biochemical Determination of the ALP Activity

Alkaline phosphatase is an enzyme expressed from osteoblast precursors and is considered one of the earliest osteogenesis-related markers. This enzyme is responsible for the hydrolysis of phosphate esters from various compounds, for the deposition of hydroxyapatite crystals in the ECM, and for the bone remodeling process. For the determination of the ALP activity, we followed a previously described protocol [45]. Briefly, the cells were washed with PBS, lysed with 100 μL of lysis buffer (0.1% Triton X-100 in 50 mM Tris-HCl, pH 10.5), and subjected to two freeze–thaw cycles of 10 min each at −20 °C and room temperature, respectively. After the lysis, an aliquot of 90 μL was mixed with 90 μL of the substrate solution (2 mg/mL p-nitrophenyl phosphate (pNPP, Sigma, Darmstadt, Germany) in 50 mM Tris-HCl and 2 mM MgCl_2_, pH 10.5) for 1 h at 37 °C. A total of 100 μL of the solution was transferred to a 96-well plate and the absorbance was measured at 405 nm with a spectrophotometer (Synergy HTX Multi-Mode Microplate Reader, BioTek, Winooski, VT, USA). The absorbance values were translated to pNP concentrations with a calibration curve of known pNP concentrations. ALP activity values were normalized to total cell number, determined previously by means of a PrestoBlue^TM^ assay. For the analysis, quadruplicates of three independent experiments were performed (*n* = 12).

### 2.8. Calcium Concentration in Supernatants

Calcium production and deposition in the ECM of osteoblasts are late markers of osteogenesis and key markers of osteoblast regulation. The calcium concentration was calculated using the O-cresol phthalein complexone (CPC) staining method (Biolabo, Maizy, France) as previously described [45]. Briefly, the cell supernatants were collected every 3 days for 21 days, and 10 μL of each supernatant was mixed with 100 μL of calcium buffer and 100 μL of the CPC calcium dye. The final solution was left at room temperature for 10 min and the absorbance was measured at 550 nm with a spectrophotometer (Synergy HTX Multi-Mode Microplate Reader, BioTek, Winooski, VT, USA). The absorbance values were translated to calcium concentration (μg/mL) using a calibration curve constructed of known calcium concentrations. Blank values of each material without cells were subtracted. All samples were analyzed in triplicates of three independent experiments (*n* = 9).

### 2.9. Collagen Production and Deposition in the ECM

Collagen type I is a fibrillar type of collagen and is the most abundant collagen that plays a key structural role in several tissues [46]. Moreover, collagen type I is the major organic component of the ECM of bone and plays a significant role in bone tissue engineering. The collagen deposition in the ECM and the determination of its concentration in direct contact with MC3T3-E1 cells with nanoHA materials were measured with Sirius Red staining with a slightly modified protocol [47]. Briefly, the cells were washed three times with PBS and fixed with 4% paraformaldehyde (PFA) for 15 min. They were washed again with PBS and stained with 0.1% Sirius Red (Sigma Aldrich, Darmstadt, Germany) in 0.5 M acetic acid for 1 h. The cells were washed again with 0.5 M acetic acid for the removal of the nonspecifically bound dye and optical microscopy photos were taken. For the quantification of collagen concentration, the bound dye was extracted with 0.5 M NaOH and the absorbance was measured at 570 nm with a spectrophotometer (Synergy HTX Multi-Mode Microplate Reader, BioTek, Winooski, VT, USA). The absorbance values were translated to collagen concentration with the use of a standard curve of known concentrations of rat tail collagen (Sigma Aldrich, Darmstadt, Germany). The collagen concentration was normalized to a cell number, previously determined with a PrestoBlue^TM^ assay. Quadruplicates of three independent experiments (*n* = 12) were analyzed.

### 2.10. RNA Isolation and Reverse Transcription Quantitative PCR (RT-qPCR)

Total RNA extraction was performed with TRIZOL reagent (Thermo Fisher Scientific, Waltham, MA, USA) according to the manufacturer’s instructions. The RNA quantity and purity were determined with UV spectrometry at 260 and 280 nm using a Nanodrop ND 100 (Thermo Fisher Scientific, Waltham, MA, USA). Complementary DNA (cDNA) was synthesized from 500 ng of RNA with the PrimeScript RT Reagent Kit (Perfect Real Time, TAKARA, Shiga, Japan) according to the manufacturer’s instructions. A quantitative real-time polymerase chain reaction (qPCR) was performed using KAPA SYBR Fast Master Mix (2×) Universal (Kapa Biosystems, Sigma, Darmstadt, Germany) in the Connect Bio-Rad real-time PCR system (Bio-Rad, Hercules, CA, USA). Four osteogenesis-related markers were chosen to be analyzed: alkaline phosphatase (ALP), bone gamma-carboxyglutamic acid-containing protein (BGLAP, osteocalcin), secreted protein acidic and rich in cysteine (SPARC, osteonectin), and secreted phosphoprotein 1 (SPP1, osteopontin). The Primer-Blast software (http://www.ncbi.nlm.nih.gov/BLAST (accessed on 10 November 2020)) was used for the primer design, as shown in Table 2. The amplification profiles for PCR were optimized for primer sets. For osteocalcin, osteonectin, and osteopontin, the real-time PCR reaction was run at 95 °C for 3 min followed by 40 amplification cycles at 95 °C for 3 s and 58 °C for 30 s. For ALP, the reaction was run at 95 °C for 3 min followed by 40 amplification cycles at 95 °C for 3 s and 60 °C for 30 s. The run was completed with the dissociation curve beginning at 65 °C for 5 s and increasing to 95 °C with 0.5 °C increments. Data were analyzed with the Bio-Rad CFX manager software version 3.0. The relative expression of the target genes was calculated using the ΔΔCq (where Cq is the threshold cycle) method after normalization to two housekeeping genes, hypoxanthine phosphoribosyltransferase-1 (Hprt) and actin, evaluated by BestKeeper. Triplicates from each time point were analyzed.

### 2.11. Statistical Analysis

All experimental results are presented as mean ± standard deviation (SD). The statistical analysis was performed using the one-way ANOVA Dunnett’s multicomparison test in GraphPad Prism version 8 software (GraphPad Software, San Diego, CA, USA) comparing each nanoHA material with the TCPS control, as well as comparing the three nanoHA compounds at each experimental time point. *p*-values indicate statistically significant differences (* *p* < 0.05, ** *p* < 0.01, *** *p* < 0.001, **** *p* < 0.0001, ***** *p* < 0.00001).

## 3. Results

### 3.1. Characterisation of the nanoHA Particles

The different nanoHA samples, with and without Sr substitution, showed particles with a rod-like shape (Figure 1). The length/width ratio appeared to increase for the fully Sr-substituted samples, whose grains were more elongated than those of pure nanoHA (Figure 2). A calculation of the presented particles (*n* = 4) in the images of Figure 2 show a length/width ratio of 2.7 nm and 3.7 nm for nanoHA and Sr-nanoHA_100, respectively. 

EDS analysis confirmed the incorporation of Sr. It has to be noted that a small amount of Ca was still measured in the Sr-nanoHA_100 (Figure 1). The (Ca + Sr)/P molar ratio was 1.83 for nanoHA, 1.65 for Sr-nanoHA_50, and 1.73 for Sr_nanoHA_100 powders. 

The XRD showed that the three nanoHA-based materials presented reflection peaks ascribable to the HA crystalline phase, based on the ICDD (International Centre for Diffraction Data) database. In detail, Figure 3 highlights that the materials presented XRD reflections characterized by different heights and widths. In particular, the pure nanoHA presented three separate peaks of diffraction at 211, 300, and 202; Sr-nanoHA_50 and Sr-nanoHA_100 presented only one and two merged peaks, respectively. 

Additionally, according to the Scherrer equation, the crystallite length for the nanoHA, the Sr-nanoHA_50, and the Sr-nanoHA_100 were 27.3 nm, 18.2 nm, and 31.1 nm, respectively, proving a disordering effect of the contemporaneous presence of Sr and Ca ions on the lattice order degree. Finally, the addition of Sr in the nanoHA caused a shift in the reflection peaks toward lower degrees 2ϑ, which was due to the increase in the unit cell parameters when Sr replaced Ca in the lattice (Figure 3). 

The cumulative release kinetics of Sr^2+^ showed that nearly 20% of Sr was released from the Sr-nanoHA_50 after 24 h and from the Sr nanoHA_100 after 3 days. At the end of the 21 days, 45% of the total amount of Sr was released from both the Sr-substituted nanoHAs. These results show that the first three days were associated with high release kinetics that then slowed down during the remaining time. In total, nearly 40 mg/L and 65 mg/L of Sr^2+^ were released from the Sr-nanoHA_50 and the Sr-nanoHA_100, respectively (Figure 4). 

### 3.2. Effect of Hydroxyapatite Materials on Cell Viability and Proliferation

In general, all the nanoHA-based materials demonstrated enhanced cell viability values with slight differences between the different compositions (Figure 5). Specifically, after 3 days in culture, Sr-nanoHA_100 had no significant differences from the TCPS control, while the other two tested materials presented decreased cell viability. However, after day 7 of culture, all materials depicted increased cell viability, with comparable values for the TCPS and the nanoHA-based materials. Moreover, it seems that there was a saturation in the cell number from the first time point; thus, no significant differences between the next time points were observed. 

Alongside the quantitative cell viability evaluation, the proliferation of pre-osteoblastic cells was assessed qualitatively following live/dead staining with calcein-AM and EthIII, respectively. The representative microscopy images are shown in Figure 6. At day 3, based on their morphology, the cells seemed to be stressed in direct culture with all materials; however, there were no visible dead cells, except for a few in the presence of Sr-nanoHA_50. However, on day 7, only living cells (depicted in green) were observed on all materials, without any dead cells (red spots). On day 14, a few dead cells were present, but these were most probably due to the saturation of the surface and consequent lack of free space to proliferate. 

### 3.3. Effect on Hydroxyapatite Materials on Osteogenesis-Related Markers

ALP activity has been evaluated as an early-stage marker of osteogenesis in MC3T3-E1 pre-osteoblastic cells in direct contact with three nanoHA-based materials at a concentration of 37.5 mg/mL of Ca_(10−x)_Sr_x_(PO_4_)_3_(OH)_2_ for 3, 7, and 14 days (Figure 7a). On days 3 and 7, all the materials depicted the same ALP profile as the TCPS control, while after 14 days in culture, Sr-nanoHA_100 exhibited significantly higher enzymatic activity compared to all the other tested conditions. On day 14, the Sr-nanoHA_100 indicated a two-fold-higher absolute released amount of Sr ions compared to Sr-nanoHA_50, similarly to day 7, as indicated by the release kinetics in Figure 4. Although the ALP activity on Sr-nanoHA_100 showed levels comparable to Sr-nanoHA_50 on day 7, it presented significantly higher levels on day 14, probably due to a cumulative effect of Sr ions that affect various intracellular signaling pathways involved in osteogenic differentiation [48], which may be in a dose-dependent manner. Moreover, ALP activity presented an increasing pattern from day 3 to day 7. Between days 7 and 14, there were no significant changes in the enzymatic activity levels. These results reveal an initiation of pre-osteoblastic maturation from day 7 of culture.

The quantification of calcium secretion levels produced by mature osteoblasts was assessed after 7, 14, and 21 days in culture of pre-osteoblasts with nanoHA-based materials, in the presence of osteogenic medium (Figure 7b). All three materials had comparable calcium concentration production at all time points, with significantly higher values compared to the TCPS control. The calcium levels were high from day 7 of culture, indicating that the cells initiated their maturation early, similarly to the ALP activity, exhibiting a correlation with the Sr release kinetics that indicated a higher release within the first three days. Interestingly, Sr-nanoHA_50 depicted lower ALP activity values and higher calcium production levels compared to Sr-nanoHA_100 after 14 days in culture. 

The collagen concentration was determined after 7, 14, and 21 days in culture. Representative optical microscopy images are shown in Figure 8a, revealing the formation of a healthy ECM in all the tested materials and the TCPS control. This is expected due to the osteoinductive properties of the nanoHA-based materials. The quantification of collagen concentration and visualization of collagen deposition revealed that all materials had significantly higher collagen production and deposition levels compared to the TCPS control at all the tested time points (Figure 8b). These results demonstrate that the three nanoHA-based compounds strongly support ECM formation and, thus, osteoblastic cell differentiation and maturation. 

The gene expression analysis of four osteogenesis-related markers was performed by means of RT-qPCR. ALP, osteonectin (OSN), osteopontin (OSP), and osteocalcin (OSC) expression was evaluated after 7 and 14 days in direct culture of nanoHA-based materials with MC3T3-E1 pre-osteoblastic cells (Figure 9). ALP (Figure 9a), a factor that plays a key role in the formation of HA crystals in the ECM, was downregulated in all the four nanoHA formulations after both the tested time points in comparison with the corresponding control, while there was an increase in the expression from day 7 to day 14 for all the materials. A comparison of the gene expression levels of ALP with the enzyme activity revealed that even if the TCPS control had significantly higher gene expression levels of ALP, the final produced protein had levels of functionality comparable to the nanoHA-based materials. These results suggest that nanoHA materials affect the translation process of the pre-osteoblastic cells. OSN expression (Figure 9b) was upregulated in nanoHA and Sr-nanoHA_50 after 7 days in culture and in all the materials after 14 days in culture in comparison with the TCPS control, while no significant differences were captured among the nanoHA-based materials. OSP expression (Figure 9c) was upregulated in all the materials, except for Sr-nanoHA_100 after 7 days in culture with a two-fold increase in comparison with the TCPS, while after 14 days in culture, the nanoHAs had expression profiles comparable to the TCPS control. Lastly, OSC gene expression, which is a late marker of osteogenesis, depicted an upregulated pattern for all nanoHA materials, with two- and three-fold-higher expression levels after 14 days in culture in comparison with the TCPS control.

## 4. Discussion

Nanosized nanoHA materials have attracted increasing interest over the years due to the formation of complex interactions with cells compared to their microsized counterparts, as they more closely mimic the naturally occurring mineral phase of bone [49,50]. For this reason, three different nanoHA materials were synthesized and their biocompatibility and osteogenic potential were evaluated in pre-osteoblastic cells. The investigated materials included pure nanoHA and nanoHA substituted with either 50 or 100% Sr ions based on its relevant use for the treatment of osteoporosis, as well as literature data on its effect on both osteoblasts and osteoclasts [24,51].

The substitution of Ca for Sr in the HA unit cell caused a distortion of the lattice and an increase in the unit lattice length due to the higher Sr ion radius (0.120 nm) compared to the Ca one (0.099 nm). This increase in the unit cell was reflected in the shift in the diffraction peaks towards lower 2ϑ degrees. The difference in the ionic radius was significant (≈13%), and when Sr and Ca ions were both contained in the unit cell (Sr-nanoHA_50), their presence exerted a negligible strain on the unit cell, affecting its order degree and leading to the observed smaller crystallites and broader diffraction peaks. This is in accordance with previous data [38,52] highlighting a decreased crystallinity for the Sr-nanoHA_50 compared to the pure and the fully substituted HA. This decreased crystallinity can also explain the higher early Sr^2+^ release kinetics for Sr-nanoHA_50. The strain developing in the lattice promotes the formation of defects within the crystalline structure that act as preferential sites for dissolution [53]. The early nature of the fast release from Sr-nanoHA_50 may be explained by the formation of a hydrated layer on the apatite grains from which Sr^2+^ can be easily released. It has been shown that the introduction of Zn, which also has a larger ion radius (0.130 nm) compared to Ca in the HA lattice, promotes the development of a hydrated layer [54]. Such hydrated layers, in which ions are very mobile, are known to be associated with a high surface reactivity [55].

The adhesion and proliferation of MC3T3-E1 pre-osteoblastic cells in direct contact with the nanoHA-based materials at a concentration of 0.25 %*v*/*v* of the product—which corresponds to 37.5 mg/mL of Ca_(10−x)_Sr_x_(PO_4_)_3_(OH)—were evaluated and compared to the tissue-culture-treated polystyrene control. The results from the viability assessment and the live/dead assay revealed that all three materials are biocompatible, without giving rise to any cytotoxic effects. This is in agreement with previous results on nanoHA materials that showed the same viability pattern at a range of tested concentrations from 0.1 to 2 %*v*/*v* [2]. 

Besides the absence of cytotoxicity, another key factor for a biomaterial to be promising for bone tissue engineering is its ability to enhance the osteogenic potential. For the biochemical evaluation of osteogenesis, ALP activity was chosen as an early marker, whereas collagen production and calcium concentration were selected as middle and late markers of osteogenesis, respectively. Considering the ALP activity data, all the materials depicted comparable levels after 3, 7, and 14 days in culture in comparison with the control culture, while only Sr-nanoHA_100 had significantly higher ALP activity after 14 days in culture. Moreover, all the tested conditions showed a 2-fold increase in the enzymatic activity from day 3 to 7 of culture, while the ALP activity retained the same levels between days 7 and 14 [56]. The calcium quantification results showed that all tested nanoHA compositions promoted osteogenic differentiation with higher calcium concentrations compared to the control at all time points [56], while the levels of the produced calcium were comparable between the different tested time points. The collagen concentration levels exhibited the same pattern as the calcium production assessment, with all materials showing significantly higher collagen values compared to the control at all time points, signifying the increased osteoinductive character of the nanoHA materials [57].

RT-qPCR experiments were performed to evaluate the osteogenic potential of nanoHA material at the gene expression level. ALP expression was suppressed in all the nanoHA samples compared to the TCPS control. This is in agreement with previous reports on nanoHA materials, suggesting that nanoHA stimulates DNA methylation and, thus, suppresses the gene expression levels of key osteogenic markers such as ALP [58]. Although ALP gene expression was significantly suppressed compared to the control, the biochemically determined ALP enzymatic activity had comparable levels for all tested conditions at day 7. This finding shows that the inhibition of ALP gene expression does not affect the functionality and regulation of the translated protein. Osteonectin, a secreted protein during the bone formation process, shows affinity and binds to collagen, initiating mineralization [59,60]. Osteonectin’s upregulated expression in all the tested nanoHA and Sr-substituted nanoHA materials validates the enhancement of the bone-forming process from the early time points of incubation with the materials. Previous work on alginate–gelatin/nanoHA microcapsules reported upregulated expression of osteonectin in dental pulp stem cells after 21 days of culture compared to cells without microcapsules, suggesting that cell maturation is affected by the presence of nanoHA in a more prolonged culture period [61]. Osteopontin, another important factor in the bone-remodeling process, plays a key role in the anchoring process of osteoclasts for the initiation of osteoclast bone resorption [62]. NanoHA materials affect the expression of osteopontin secreted for the initiation of bone resorption only after 7 days of culture, suppressing its expression after 14 days of culture, thus hindering the anchoring and activity of osteoclasts. The upregulation after 7 days of culture is in agreement with a previous study on Sr-substituted nanoHA in fibroin/carboxymethylated chitosan [32]. Lastly, osteocalcin, a bone marker of the mature stages of osteogenesis [63], showed a significant gene expression profile after 14 days of culture at all the tested nanoHA materials, indicating that after two weeks, there are fully formed osteoblasts with enhanced activity compared to the control. These results are in agreement with previously reported results on nanoHA materials in indirect culture with endothelial mesenchymal stem cells for 7, 14, and 21 days, showing an upregulation of osteocalcin expression after 14 days of culture [56]. A previous study on Sr-substituted silk fibroin/carboxymethylated chitosan scaffolds showed increased osteocalcin expression after 14 days of culture compared to the control scaffold without Sr [32]. 

Comparing the osteogenic differentiation assessment of pre-osteoblastic cells in direct contact with the two Sr-substituted nanoHAs, Sr-nanoHA_50 and Sr-nanoHA_100, the data did not reveal significant variations for each osteogenic marker at any investigated time point. This observation may be explained by the release profile of Sr ions, which showed a difference between the two materials only for the first few days, while 45% of the total Sr amount was released after 21 days from both Sr-substituted nanoHAs. The fact that the first three days were associated with a high Sr release that was then slowed down may explain the observation of an early osteogenic maturation of cells, as evidenced by the calcium and ALP activity data.

The synthesized and osteogenically evaluated nanoHA-based materials are excellent osteoinductive materials that can be incorporated within other material types and 3D scaffolds and devices for the enhanced treatment of bone diseases.

## 5. Conclusions

In this study, we investigated the biological responses of pre-osteoblasts treated with three nanoHA-based materials: nanoHA and two different substitution degrees of Ca with Sr, 50 and 100%, respectively. All materials featured needle-shaped crystals and Sr ions were released upon material dissolution, showing that the concomitant presence of both Sr and Ca ions led to a less ordered structure more prone to dissolution. None of the nanoHA materials revealed any cytotoxic effects at a concentration of 0.25 %*v*/*v* of the product, which corresponds to 37.5 mg/mL of Ca_(10−x)_Sr_x_(PO_4_)_3_(OH), at all tested time points. All three materials enhanced the osteogenic differentiation of pre-osteoblasts in direct contact, as evidenced from the ALP activity, calcium and collagen production, and gene expression data. The Sr-nanoHA_100 indicated a significant increase in the ALP activity on day 14 compared to the control and the other two nanoHA-based compounds. Both Sr-substituted nanoHAs enhanced the gene expression of ALP after 7 days and OSC after 14 days compared to the pure nanoHA. No significant differences for each osteogenic marker at all the investigated time points were observed between the Sr-nanoHA_50 and Sr-nanoHA_100, which may be explained by the Sr release kinetics. An early osteogenic maturation of the pre-osteoblasts may rely on the high Sr release rate on the first three days, which was then slowed down until 21 days. Overall, the data confirm the great osteogenic potential of the produced nanoHA and Sr-substituted nanoHA compounds, which may be exploited in combination with other materials for scaffold fabrication in bone tissue engineering. 

## Figures and Tables

**Figure 1 nanomaterials-13-01881-f001:**
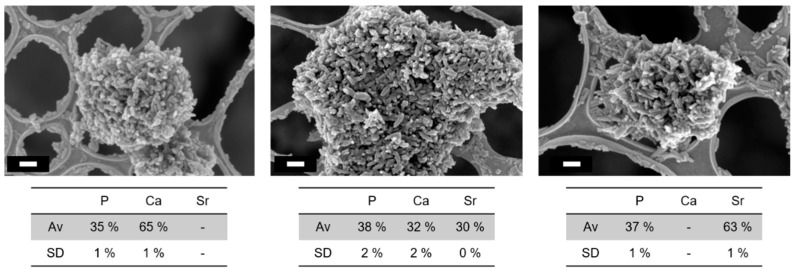
FE-SEM images and EDS atomic percentage (average and standard deviation) of the nanoHA (**left**), the Sr-nanoHA_50 (**middle**), and the Sr-nanoHA_100 (**right**). Scale bar = 100 nm. The atomic percentages were measured on three different areas of each sample (120 µm × 120 µm).

**Figure 2 nanomaterials-13-01881-f002:**
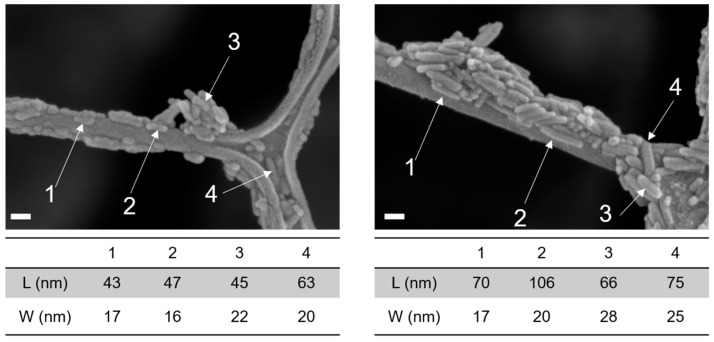
Comparison of length/width aspect ratio of nanoHA (**left**) and Sr-nanoHA_100 (**right**). Scale bar = 40 nm.

**Figure 3 nanomaterials-13-01881-f003:**
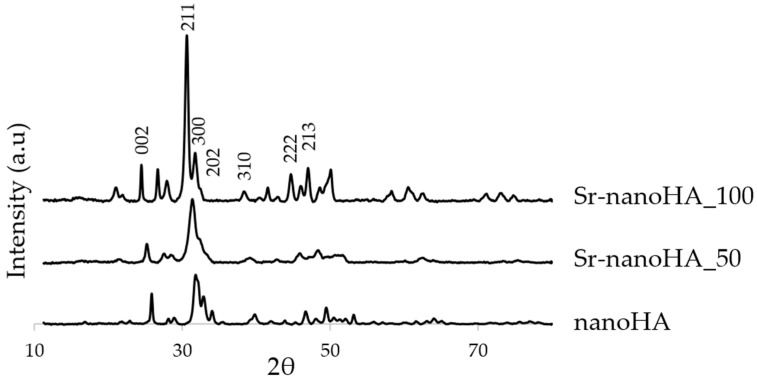
XRD diffractograms of the nanoHA and the substituted nanoHAs with 50% (Sr-nanoHA_50) and 100% (Sr-nanoHA_100) Sr.

**Figure 4 nanomaterials-13-01881-f004:**
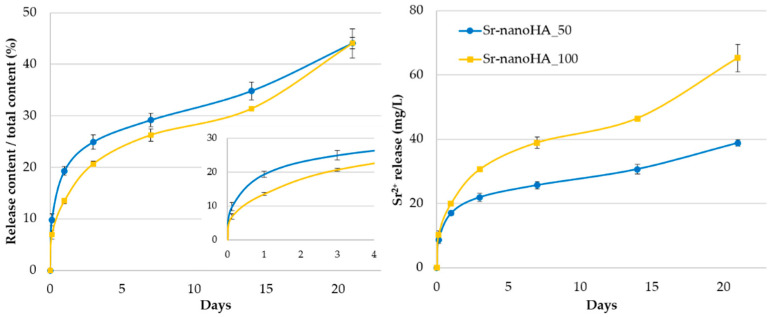
Sr cumulative release kinetics from Sr-nanoHAs. (**left**) ratio between the release content and the initial total content; inset: magnification of this ratio during the first 4 days; (**right**) absolute released amount.

**Figure 5 nanomaterials-13-01881-f005:**
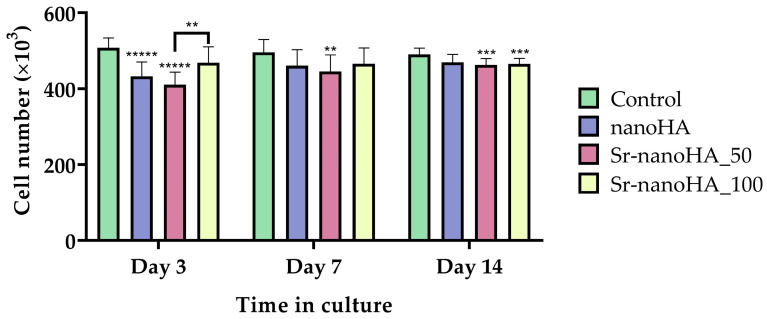
Cell viability assessment of nanoHA materials in direct culture with MC3T3-E1 pre-osteoblastic cells after 3, 7, and 14 days in culture. Each bar represents the mean ± SD of quadruplicates of three independent experiments (*n* = 12). Statistical analysis was performed for each material compared to the TCPS control and among the three nanoHA-based materials at the corresponding time point (** *p* < 0.01, *** *p* < 0.001, ***** *p* < 0.00001).

**Figure 6 nanomaterials-13-01881-f006:**
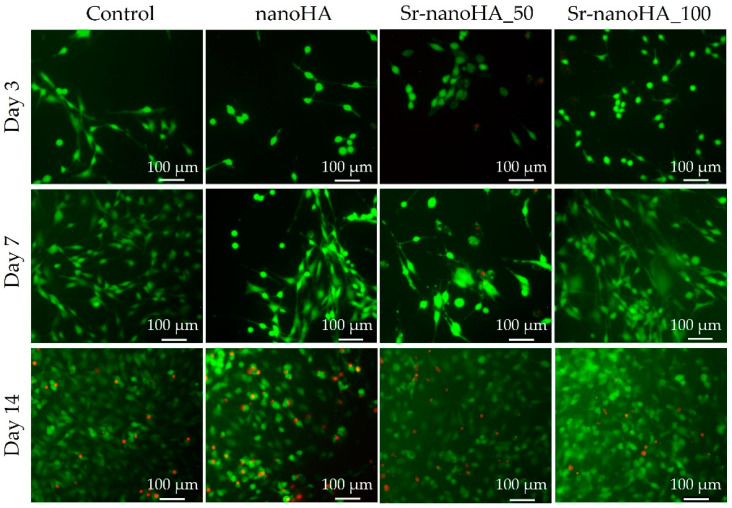
Live/dead assay images showing the viability of pre-osteoblastic cells cultured in direct culture with nanoHA materials and TCPS control after 3, 7, and 14 days. Scale bars represent 100 μm.

**Figure 7 nanomaterials-13-01881-f007:**
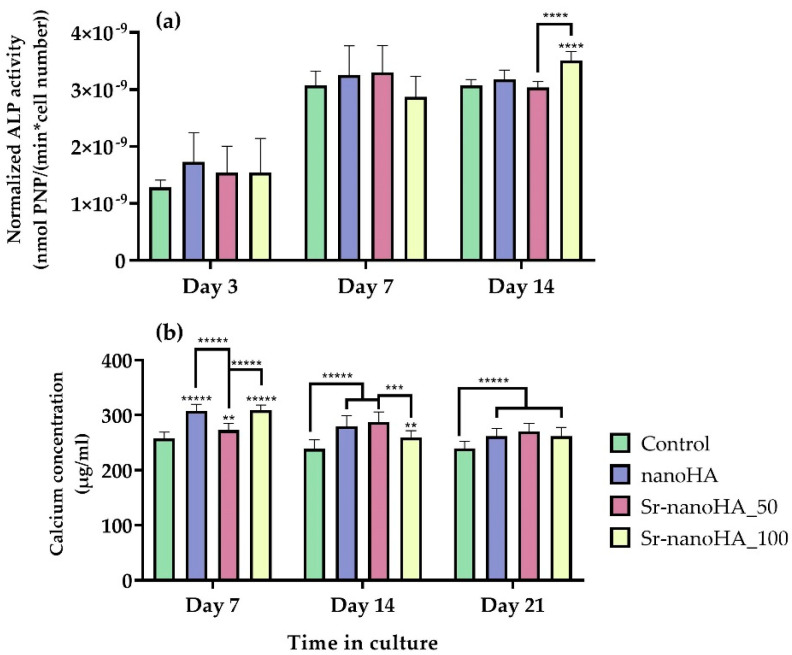
Osteogenic potential of cells in direct contact with nanoHA materials. (**a**) Normalized ALP activity with cell number after 3, 7, and 14 days of culture; (**b**) calcium concentration produced by cells cultured for 7, 14, and 21 days. Each bar represents the mean ± SD of quadruplicates of three independent experiments (*n* = 12). Statistical analysis was performed for each material compared to the TCPS control and among each other at the corresponding time point (** *p* < 0.01, ****p* < 0.001, **** *p* < 0.0001, ***** *p* < 0.00001).

**Figure 8 nanomaterials-13-01881-f008:**
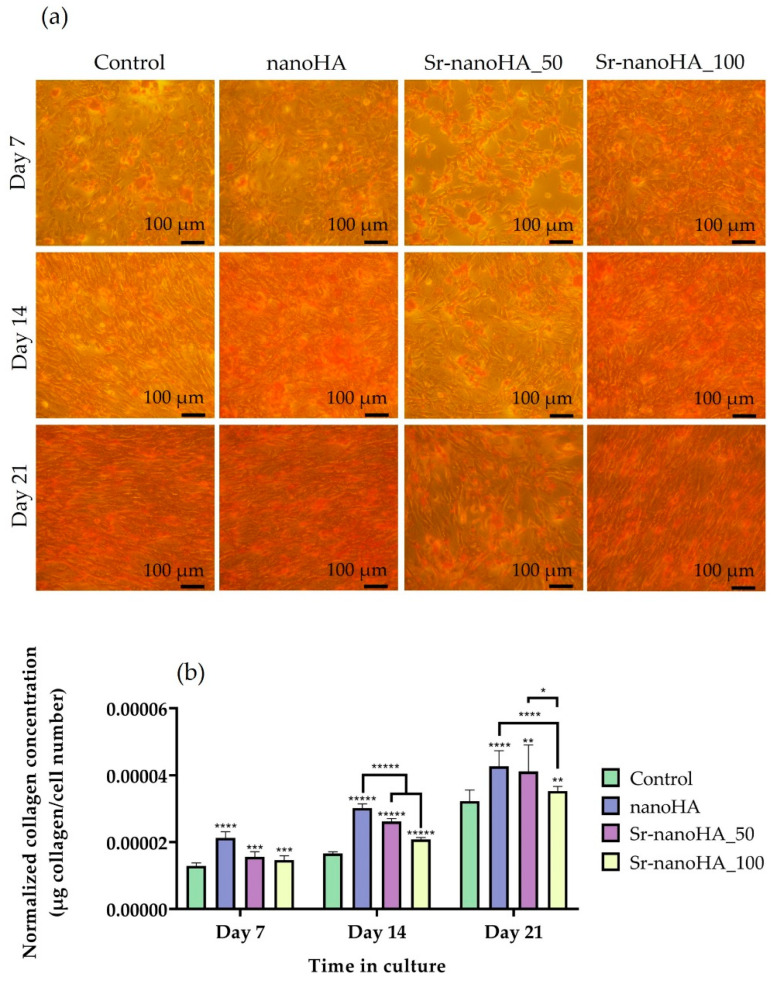
Collagen production and deposition on the ECM of pre-osteoblastic cells on TCPS and hydroxyapatite materials after 7, 14, and 21 days: (**a**) Microphotographs of collagen deposition stained with Sirius Red dye. Scale bars represent 100 μm; (**b**) quantitative results of collagen production after extraction with NaOH normalized to cell number. Each bar represents the mean ± SD of quadruplicates of three independent experiments (*n* = 12). Statistical analysis was performed for each material compared to the TCPS control and between each other at the corresponding time point (* *p* < 0.05, ** *p* < 0.01, *** *p* < 0.001, **** *p* < 0.0001, ***** *p* < 0.00001).

**Figure 9 nanomaterials-13-01881-f009:**
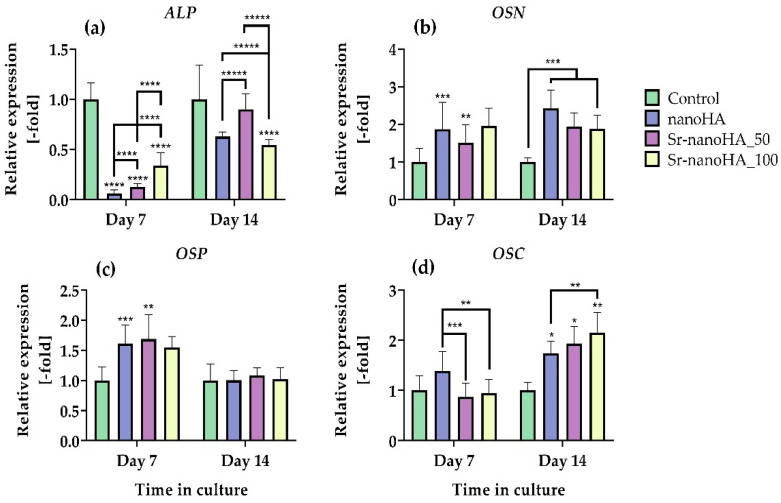
Quantitative real-time PCR of osteogenic gene expression markers of MC3T3-E1 pre-osteoblastic cells cultured for 7 and 14 days in direct culture with nanoHA materials in relative quantification with the TCPS control. ALP (**a**), osteonectin (OSN) (**b**), osteopontin (OSP) (**c**), and osteocalcin (OSC) (**d**) relative gene expression levels were assessed with qPCR after 7 and 14 days of culture. Relative gene expression levels were analyzed using the ΔΔCt method by normalizing with two housekeeping genes (HPRT and actin) as endogenous controls. Each bar represents the mean ± SD of triplicates of three independent experiments (*n* = 9). Statistical analysis was performed for each material compared to the TCPS control and among the three nanoHA-based materials at the corresponding time point (* *p* < 0.05, ** *p* < 0.01, *** *p* < 0.001, **** *p* < 0.0001, ***** *p* < 0.00001).

**Table 1 nanomaterials-13-01881-t001:** Acronyms of the produced nanoHA-based materials.

Sample Acronym	Substituted Element	Degree of Substitution (%mol)	Composition
nanoHA	-	-	Ca_10_ (PO_4_)_6_ (OH)_2_
Sr-nanoHA_50	Sr	50	Ca_5_Sr_5_(PO_4_)_6_ (OH)_2_
Sr-nanoHA_100	Sr	100	Sr_10_ (PO_4_)_6_ (OH)_2_

**Table 2 nanomaterials-13-01881-t002:** Primers designed for RT-q-PCR.

Gene	Forward (5′-3′)	Reverse (5′-3′)
Actin (housekeeping)	TCGTGTTGGATTCTGGGGAC	ACGAAGGAATAGCCACGCTC
HPRT (housekeeping)	TGGGCTTACCTCACTGCTTT	ATCGCTAATCACGACGCTGG
ALP	TGCCTACTTGTGTGGCGTGAA	TCACCCGAGTGGTAGTCACAATG
SPP1 (osteopontin)	CCTGGCTGAATTCTGAGGGAC	TATAGGATCTGGGTGCAGGCT
BGLAP (osteocalcin)	ACCATCTTTCTGCTCACTCTGC	CTTATTGCCCTCCTGCTTGGA
SPARC (osteonectin)	CGGACCATGCAAATACATCGC	CTCAAAGTCTCGGGCCAACA

## Data Availability

The data presented in this study are openly available in ZENODO: https://doi.org/10.5281/zenodo.7589462 (accessed on 31 January 2023).

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
