# Peer review of "Osteogenic Potential of Nano-Hydroxyapatite and Strontium-Substituted Nano-Hydroxyapatite"

_nanomaterials, 2023, doi:10.3390/nano13121881_

Round 1
Reviewer 1 Report
Presented paper is devoted to the study of osteogenic potential of nanosized HA containing Sr. The paper seems interesting; however, there are several comments to it.
1. Introduction. Please put more attention to how HA NP can be used.
Lines 35, 36. Please support this sentence with some references. For example, https://doi.org/10.1016/j.jksus.2023.102681, https://doi.org/10.1016/j.matlet.2022.132099, https://doi.org/10.1016/j.kjs.2023.02.014.
Line 112. What was the source of Sr?
Lines 143, 144. Why was Tris HCl used as immersion medium? Usually, the solutions more close to the human fluids is used during release studies.
Line 295. Please separate sentences.
Lines 301–304. EDS data (maps or table) should be provided.
Fig. 1. Please make scale bars more distinguishable.
Fig. 2. The length/width data are undistinguishable.
Fig. 3. Since there are several patterns on one fig., please, delete “Intensity (a.u.)”.
Line 327. Please indicate that release is cumulative.
Fig. 4. Please describe why at day 7 Sr-nanoHA_100 demonstrated less ALP activity (in comparison with other samples) but at day 14 is vice versa.
Author Response
Reviewer No. 1
Presented paper is devoted to the study of osteogenic potential of nanosized HA containing Sr. The paper seems interesting; however, there are several comments to it.
- Introduction. Please put more attention to how HA NP can be used.
Response: We thank the reviewer for the opportunity that gave us to highlight the use of HA NPs in a wide variety of applications in form of injectable medical devices, 3D composite scaffolds and coatings on metallic bone implants. The additions in the Introduction of the manuscript can be seen in lines 48-57 and 65-68: “Specifically … recently reported [13]”, lines 49-53, “Moreover, nanoHA holds great promise … and bone infections [21]”.
Lines 35, 36. Please support this sentence with some references. For example, https://doi.org/10.1016/j.jksus.2023.102681, https://doi.org/10.1016/j.matlet.2022.132099,
https://doi.org/10.1016/j.kjs.2023.02.014.
Response: We thank the reviewer for this comment. The suggested publications have been thoroughly inspected, and we have decided that the following study can be a great supplement for our manuscript: “High performance of coating hydroxyapatite layer on 316L stainless steel using ultrasonically and alkaline pretreatment” (ref 4 in the manuscript, line 44). We therefore added this reference. The other references were not close to the subject of this work.
Line 112. What was the source of Sr?
Response: We thank the reviewer for this comment. The source for Sr2+ was SrCl2.H2O. We added this information in the text (lines 131-133): “The synthesis of Sr-nano-HA protocol consisted in feeding a solution containing calcium (CaCl2)/ strontium (SrCl2.6H2O) and a phosphate (KJ2PO4) solution.”
Lines 143, 144. Why was Tris HCl used as immersion medium? Usually, the solutions more close to the human fluids is used during release studies.
Response: The authors agree with the reviewer’s comment since SBF or PBS are often used in the literature as media mimicking the physiological fluids for conducting the release tests. However, the use of these media has been proved to highly affect the release kinetics mostly due to the high ion concentration and the occurrence of precipitation reactions at the surface of the investigated materials. Due to the variability of the obtained results, several authors have proposed TRIS-HCl buffer as an alternative release medium as it allows to achieve more reliable results.
In the present contribution, the same protocol described in a previous contribution by the authors on strontium containing materials (The Incorporation of Strontium to Improve Bone-Regeneration Ability of Mesoporous Bioactive Glasses Materials 2018, 11(5), 678) has been used. The employed procedure has been optimized by Shi et al. in the following reference “Copper-doped mesoporous silica nanospheres, a promising immunomodulatory agent for inducing osteogenesis” Acta Biomater. 2016, 30, 334–344.
Line 295. Please separate sentences.
Response: We thank the reviewer for the correction. The sentences have been separated.
Lines 301–304. EDS data (maps or table) should be provided.
Response: Figure 1 has been modified. The EDS atomic percentage table has been added. The captions have been modified accordingly.
Fig. 1. Please make scale bars more distinguishable.
Response: Scales have been made more distinguishable in Figure 1.
Fig. 2. The length/width data are undistinguishable.
Response: The length/width data have been made more distinguishable in Figure 2.
Fig. 3. Since there are several patterns on one fig., please, delete “Intensity (a.u.)”.
Response: A complete graph should show the axis title, we therefore did not delete it.
Line 327. Please indicate that release is cumulative.
Response: This has been modified in the text and in the caption accordingly by adding it in line 350 and Figure 4 caption.
Fig. 4. Please describe why at day 7 Sr-nanoHA_100 demonstrated less ALP activity (in comparison with other samples) but at day 14 is vice versa.
Response: We thank the reviewer for this comment. On day 7 there are no significant differences among the tested materials, while on day 14 the ALP activity of Sr-nanoHA_100 is significantly higher in comparison with the Sr-nanoHA_50.
Although the release kinetics of both Sr-substituted nanoHAs show an almost two-fold absolute released amount of Sr from Sr-nanoHA_100 compared to Sr-nanoHA_50 after 7 days, and we expect an increase in cell differentiation, the ALP activity values are comparable. A possible explanation for this observation may include the concentration of the strontium ions that are known to affect various intracellular signaling pathways involved in the osteogenic differentiation in a dose-dependent manner (https://doi.org/10.1016/8756-3282(95)00484-X). We added the text in lines 396-402 “On day 14, … dose effects.”

Reviewer 2 Report
In this study, the authors investigated the biological responses of pre-osteoblasts treated with three nanoHA based materials, nanoHA and two different substitution degrees of Ca with Sr by 50 and 100%, respectively. All materials featured needle shape crystals and Sr ions were released upon material dissolution showing that the concomitant presence of both Sr and Ca ions led to a less ordered structure more prone to dissolution. The data confirm the great osteogenic potential of the produced nanoHA and Sr substituted nanoHA compounds, which may be exploited in combination with other materials for scaffold fabrication in bone tissue engineering. This work holds great potential in treating bone diseases and may be publishable in this journal after addressing some minor issues as follows:
1. Strontium ion substituted nHA not only possessed excellent cell compatibility but also effectively enhanced osteogenic activity, which could be used for scaffolds manufacturing in bone tissue engineering. However, the use of osteogenic scaffolds often requires surgery, so is there a non-invasive method for using this material in the body.
2. Recently, some osteogenic scaffolds with anti-tumor functions have been reported. Please cite relevant paper(Nanomaterials 2023, 13(3), 551; https://doi.org/10.3390/nano13030551).
Author Response
Reviewer No. 2
In this study, the authors investigated the biological responses of pre-osteoblasts treated with three nanoHA based materials, nanoHA and two different substitution degrees of Ca with Sr by 50 and 100%, respectively. All materials featured needle shape crystals and Sr ions were released upon material dissolution showing that the concomitant presence of both Sr and Ca ions led to a less ordered structure more prone to dissolution. The data confirm the great osteogenic potential of the produced nanoHA and Sr substituted nanoHA compounds, which may be exploited in combination with other materials for scaffold fabrication in bone tissue engineering. This work holds great potential in treating bone diseases and may be publishable in this journal after addressing some minor issues as follows:
- Strontium ion substituted nHA not only possessed excellent cell compatibility but also effectively enhanced osteogenic activity, which could be used for scaffolds manufacturing in bone tissue engineering. However, the use of osteogenic scaffolds often requires surgery, so is there a non-invasive method for using this material in the body.
Response: We thank the reviewer for raising this issue. Strontium ion-substituted nano hydroxyapatite (Sr-nanoHA) has demonstrated excellent cell compatibility and enhanced osteogenic activity. However, the conventional approach for using such osteogenic scaffolds typically involves surgical implantation eg of 3D osteogenic polymer composite scaffolds (recently reported by our group in https://www.mdpi.com/2073-4360/15/4/1052). Nevertheless, researchers have explored non-invasive methods for delivering different Sr-nanoHA, such as injectable fillers. By formulating Sr-nanoHA into biocompatible hydrogels or biodegradable polymers, it is possible to directly inject the material into the desired location. The fillers provide a scaffold-like structure that supports cell attachment, proliferation, and tissue regeneration. The Sr-nanoHA within the filler can enhance the osteogenic activity and promote bone formation at the injection site. We have added a relevant sentence and a reference suggested by the reviewer (in Nanomaterials, 2023) in the Introduction in lines 55-57 ‘In this context, … recently reported’ highlighting the use of minimally and non-invasive osteoinductive materials [13].
- Recently, some osteogenic scaffolds with anti-tumor functions have been reported. Please cite relevant paper (Nanomaterials 2023, 13(3), 551; https://doi.org/10.3390/nano13030551).
Response: We thank the reviewer for this suggestion. We have added it to our manuscript (lines 55-57) as it presents an excellent review on bifunctional scaffolds for minimally invasive photothermal therapeutic agents towards bone tumors ablation and bone regeneration

Round 2
Reviewer 1 Report
The authors carefully corrected all comments. I suppose that the work can be accepted.